# *In Vitro* Investigation of the Cytotoxic and Antiproliferative Effects of *Haberlea rhodopensis* Total Extract: A Comparative Study

Martina I. Peeva [1], Maya G. Georgieva [1], Aneliya A. Balacheva [1], Atanas Pavlov [2,3,*] and Nikolay T. Tzvetkov [1]

1   Department of Biochemical Pharmacology & Drug Design, Institute of Molecular Biology "Roumen Tsanev" Bulgarian Academy of Sciences (BAS), Acad. G. Bonchev Str., bl. 21, 1113 Sofia, Bulgaria; martina.iv.peeva@gmail.com (M.I.P.); mgeorgieva@bio21.bas.bg (M.G.G.); neli_bal@abv.bg (A.A.B.); ntzvetkov@bio21.bas.bg (N.T.T.)
2   Faculty of Technologies, University of Food Technologies, 26 Maritza Blvd., 4002 Plovdiv, Bulgaria
3   R&D Department, Innova BM Ltd., 12 B, Stefan Karadja Str., 1000 Sofia, Bulgaria
*   Correspondence: at_pavlov@yahoo.com or pavlov@innovabm.com

**Abstract:** *Haberlea rhodopensis* Friv., known also as Rhodope silivryak and the Orpheus flower, is a Balkan endemic "resurrecting" plant belonging to the *Gesneriaceae* family. In folk medicine, the leaves of *Haberlea rhodopensis* Friv. were widely used to treat wounds and some infectious diseases of stock such as foot-and-mouth disease and hoof rot, while the herb of *Haberlea rhodopensis* Friv. is still used to cleanse the stomach, liver, kidneys, and blood vessels. Because of the content of myconoside, during the last decade, *Haberlea rhodopensis* Friv. extracts have been recognized as valuable cosmetic ingredients. In the present study, we aim to (i) evaluate the cytotoxic and antiproliferative activity of two herb extracts of *Haberlea rhodopensis* Friv. that are commercially used for the preparation of cosmetic ingredients on different cancer cells, with one normal cell line used as a reference, and (ii) compare the investigated effects with those observed for the reference anticancer, non-selective compound doxorubicin. Herein, we observed a decrease in the inhibitory activity of both extracts compared to those of doxorubicin against all tested cell lines. However, the myconoside-enriched *Haberlea rhodopensis* Friv. plant Extract 2 (designated also as M2) showed increased inhibitory activity (cytotoxicity and antiproliferative effects) compared to the *Haberlea rhodopensis* Friv. plant Extract 1 (designated also as E1). Moreover, the *Haberlea rhodopensis* Friv. plant Extract 2 showed a significant increase in cytotoxicity (at 24 h) and antiproliferative activity (at 48 and 72 h post-treatment) at its highest-tested concentration of 100 μg/mL compared to *Haberlea rhodopensis* Friv. plant Extract 1.

**Keywords:** antiproliferation; cisplatin; cytotoxicity; doxorubicin; flavonoid antioxidants; *Haberlea rhodopensis* Friv.





## 1. Introduction

*Haberlea rhodopensis* Friv. (HR) is a plant of the *Gesneriaceae* family with typical lilac flowers that grows in stony areas and is endemic in the Rhodope Mountains of the Thracian regions of Bulgaria and Greece (Figure S1A). *H. rhodopensis* Friv. belongs to a group of resurrection plants able to withstand prolonged drought periods, tolerating desiccation and quickly resuming growth upon rehydration [1,2]. Due to its growth in limited places, HR is a protected species in Greece and Bulgaria [3].

Primary investigations of Dell'Acqua and Schweikert proved skin benefits of HR extracts that are myconoside-rich [4]. In recent years, HR has been intensively studied in terms of its broad antimigratory and anticancer effects, antioxidant and free radical-scavenging activity, radioprotective and immunostimulatory effects, as well as antibacterial and anti-aging efficacy [5]. Therefore, the attention of researchers is focused on the investigation of the potential applications of HR in phytotherapy, human and veterinary medicine,

and cosmetics [5,6]. Extensive investigations showed that different HR extracts may contain high levels of flavonoid antioxidants, including phenolic acids (e.g., ferulic, staric, caffeic, *p*-coumaric, sinapinic acid, and others), flavonoid aglycones, and glycosides (such as luteolin, hesperidin, quercetin, myricitin, rutin, and others) [7–9]. However, spectroscopic analyses using (U)HPLC-MS showed that two glycosides—myconoside and paucifloside—together with three other hispidulin-flavone C-glycopyranosides, represent the main constituents of different ethanolic *H. rhodopensis* Friv. extracts (HREs) [5]. The three new flavone C-glycosides are hispidulin 8-C-(2-O-syringoyl glucopyranoside), hispidulin 8-C-(6-O-acetyl glucopyranoside), and hispidulin 8-C-(6-O-acetyl-2-O-syringoyl gluco-pyranoside), which were also isolated and reported by another scientific group [9]. Noteworthy, alcohol extracts from HR were investigated to show antioxidant, antiviral, antibacterial, and antifungal activities [10]. In addition, the potent antioxidant and hepatoprotective effects of myconoside were recently demonstrated. The chemical structures and IUPAC names of myconoside and paucifloside are presented in Figure S1B.

Recent experiments show that among the components with biological activity found in HR, myconoside has potent antioxidant and hepatoprotective effects [11]. Together with the phenolic content, these compounds significantly contribute to the antioxidant properties of HREs [7]. In particular, ethanolic extracts of HR leaves have been reported to exhibit vital antimicrobial and antioxidant activities, reduce the clastogenic effect of γ-irradiation, and exert in vivo anticlastogenic and antimutagenic potential against the anticancer drug cyclophosphamide [12–15].

Phenolic acids accumulated in high amounts in the resurrection plants possess therapeutic properties due to their ability to capture free radicals and decrease oxidative stress [9]. Berkov and colleagues reported their study on the polar and apolar fractions of methanol HREs by gas chromatography-mass spectrometry and they have identified five free phenolic acids, namely syringic, vanillic, caffeic, dihydrocaffeic, and *p*-coumaric [16]. Furthermore, another study demonstrated that in alcohol HRE, the most abundant phenolic acids were the sinapic, ferulic, caffeic, and p-coumaric acids, as well as at least five other phenolic acids, which, although not so abundant, were still present in the extracts [7].

HR is a rare and threatened plant and, therefore, its harvesting from nature is prohibited. In vitro cultivation of plants is a promising technology for the sustainable production of bioactive plant metabolites. On this basis, Bulgarian company Innova BM Ltd. developed in vitro cultures of HR with different degrees of differentiation that are used for industrial sources of water–alcoholic extracts [17]. These extracts are the subject of the current investigation.

In the present study, we aim to (i) evaluate the cytotoxic and antiproliferative effects of two herb extracts of HR on different cancer cell lines, with one normal cell line used as a reference, and (ii) compare the investigated effects with those observed for the reference anticancer, non-selective compound doxorubicin.

## 2. Materials and Methods

### 2.1. Haberlea rhodopensis Friv. In Vitro Culture Extracts

*Haberlea rhodopensis* Friv. in vitro culture extracts were provided by Innova BM Ltd., Bulgaria https://innovabm.com/ (accessed on 30 January 2020). The extracts were produced by the company's proprietary technology by ethanol extraction of *Haberlea rhodopensis* Friv. in vitro culture biomass [17]. The myconoside content was 115 mg/g dry extract and 215 mg/g dry extract for E1 and M2 extracts, respectively.

### 2.2. Preparation of Stock Solutions

Two extracts (indicated throughout as Extract 1 or E1 at 5.0 mg/mL in ddH$_2$O stock, and myconoside-enriched Extract 2 or M2 at 10.0 mg/mL in ddH$_2$O stock) were tested for their cytotoxic and antiproliferative effects, respectively, after 24-, 48- and 72-h treatment of three cancer and one non-cancer (reference) cell lines. The stock solutions of both HR extracts were stored at –20 °C in the dark and tempered before use. The experiments

were carried out on triple-negative, epithelial human breast adenocarcinoma (MDA-MB-231; ATCC; Manassas, VA, USA), human colorectal adenocarcinoma (HT-29; ATCC, USA), human hepatocellular carcinoma (HepG2; ATCC, USA), and mouse embryonic fibroblasts (3T3/L1; ATCC, USA) cells. Cells were incubated with serial dilutions (0.01–100 µg/mL) of both extracts using a modified MTT assay [18,19]. The anticancer, nonselective agent doxorubicin (Key Organics Ltd., Camelford, UK) (100 mM stock in DMSO) was tested in the concentration range of 0.001–100 µM (DMSO concentration $\leq$ 0.05% $v/v$) and was included in each experiment as a reference substance. For the dissolution of the insoluble formazan, a lysis solution containing N,N-dimethylsulfoxide (DMSO, Sigma-Aldrich, St. Louis, MO, USA) was used. The tests were performed in triplicate (eight replications per test) and the results presented as the mean % of the untreated controls $\pm$ SD from three independent experiments ($n$ = 3).

### 2.3. Preparation of Cell Cultures

Human breast adenocarcinoma (MDA-MB-231), human colorectal adenocarcinoma (HT-29), human hepatocellular carcinoma (HepG2), and mouse embryonic fibroblasts (3T3/L1, reference non-cancer cells) cells were cultured in Dulbecco's modified Eagle's medium (DMEM, Gibco, Vienna, Austria) in the presence of 10% fetal bovine serum (FBS; Gibco, Austria), penicillin (100 U/mL) and streptomycin (0.1 mg/mL) solution (Gibco, USA). All cells were cultivated under an atmosphere of $CO_2$ (5.0%) at 37 °C and passaged by trypsinization with trypsin-EDTA (Greiner, Pleidelsheim, Germany) at a confluence of approximately 80%. The experiments were performed with cells in the exponential phase of growth (at a density of 5000 cells/well) using 96-well flat-bottom plates at a final volume of 100 µL/well. Cells were incubated overnight before the addition of doxorubicin and/or tested extracts.

### 2.4. Determination of Cell Viability

The cytotoxicity and antiproliferation of doxorubicin (1.0 mM stock solution in DMSO), Extract 1 (E1, 10.0 mg/mL stock solution in dd$H_2O$), and Extract 2 (M2, 5.0 mg/mL, stock solution in dd$H_2O$) were assessed in HepG2, MDA-MB-231, and HT-29 cancer cell lines, as well as in 3T3 mouse embryonic fibroblasts cells used as a reference cell line by colorimetric assay, applying 3-(4,5-dimethylthiazol-2-yl)-2,5-diphenyltetrazolium bromide (MTT) (VWR, Darmstadt, Germany), as previously reported [20].

The MTT test allows for determining the linear dependency between metabolically active cells (viable cells) and the measured color intensity of the purple-colored formazan solution, which can be quantified by spectrophotometric measuring at a certain wavelength (usually 550 or 570 nm). The obtained information is then used to assess the change in cells, e.g., death and/or proliferation. The loss of intensity of the purple color is directly associated with the loss of viable cells in the presence of a cytotoxic compound (agent). The quantity of in situ-formed formazan product can be determined spectrophotometrically by the measurement of the absorption/optical density (Amax/DO), after solubilization with a lysis solution (usually with DMSO, Sigma-Aldrich, St. Louis, MO, USA). The measured OD/Amax corresponds to the number (in %) of viable cells after a certain incubation period (24, 48 or 72 h) with the tested substance.

### 2.5. MTT Assay

The MTT assay was performed with slight modifications according to the literature [18,20]. Cells were seeded in 96-well culture plates with a density of $5 \times 10^3$ cells per well. Following adherence, the cells were treated with test extracts (0.01, 0.1, 1.0, 5.0, 10, 25, 50, and 100 µg/mL) and the reference compound doxorubicin (0.01, 0.1, 1.0, 2.5, 5.0, 10, 25, 50, and 100 µM), and further incubated for 24, 48 or 72 h at 37 °C (under 5.0% $CO_2$ atmosphere). After the respective incubation period, 10 µL MTT solution (5.0 mg/mL) per well was added and cells were incubated for a further 180 min. Then, the medium was removed and a lysing solution containing DMSO was added to each

well. The plates were then shaken at room temperature until the completed dissolution of the purple crystalline product (formazan). The quantification of the produced formazans after the reduction of MTT was monitored using a microplate ELISA reader VarioscanTM LUX (Thermo Fisher Scientific Inc., Waltham, MA, USA) at 550 and 570 nm. The determined cytotoxicity/antiproliferation is expressed as percentage cell viability using the following equation:

$$\%\text{Cell viability} = (\text{ODsample} - \text{ODblank})/(\text{ODcontrol} - \text{ODblank}) \times 100 \qquad (1)$$

In this equation, OD sample, OD blank and OD control are the measured absorption of the test, blank, and control sample. The results were presented as the mean % of the untreated controls $\pm$ SD from three independent experiments ($n = 3$).

*2.6. Statistical Analysis*

The statistical analysis and the representative graphs thereof were carried out using GraphPad Prism 6.0 (GraphPad Software, La Jolla, CA, USA). The respective half maximal inhibitory concentration ($IC_{50}$) values were obtained by non-linear regression analysis. The inhibitory curves were built using the log[inhibitor] vs. normalized response—Variable slope equation with least squares fit. Since the measured inhibitory concentration (growth inhibitory activity) of the tested Extract 1 and 2 is given in µg/mL, the $IC_{50}$ values of doxorubicin (initially obtained as µM) were further converted into their respective µg/mL units using the following equation:

$$\text{µg/mL} = \text{µM} \times \text{MWdoxo (g/mol or µg/µmol)}/1000 \qquad (2)$$

For further simplification, the inhibitory activity ($IC_{50}$ values) of doxorubicin is given in µg/mL in order to better compare the respective inhibitory activity for both tested extracts (see Table S1). The experimentally estimated $IC_{50}$ values are obtained from at least three independent experiments and given with their standard deviation ($IC_{50} \pm$ SD). For the one-way ANOVA test, $p$ values $\leq 0.05$ were considered as statistically significant.

## 3. Results

In order to evaluate cell viability, a modified MTT assay was used [19]. This method is one of the most used to evaluate the viability of different cancer and non-cancer cells in the presence of tested (screened) new compounds (naturally occurring and synthetic molecules). The MTT assay is a colorimetric test based on an enzyme–catalytic reduction of 3-(4,5-dimethylthiazol-2-yl)-2,5-diphenyltetrazolium bromide (MTT) (a yellow tetrazolium salt dye) by Nicotinamide adenine dinucleotide phosphate (NAD(P)H) to its insoluble purple crystalline form formazan. The assay is used for indirectly assessing cell viability based on the capacity of viable cells to reduce the tetrazolium dye MTT in the presence of mitochondrial NAD(P)H-dependent cellular oxidoreductase enzymes [19]. Therefore, the MTT test can also be used to measure cytotoxicity (loss of viable cells), usually after 24 h treatment with an investigated substance, or cytostatic activity (antiproliferation), usually after 48 and/or 72 h treatment with a compound of interest.

*3.1. Growth Inhibitory Activity of E1 and M2 against MDA-MB-231 Cancer Cell Line*

Both extracts were first tested for their ability to inhibit the growth of triple-negative, epithelial human breast adenocarcinoma (MDA-MB-231) cell line. The MTT tests were performed in triplicate and, in each experiment, doxorubicin was used as a reference compound. The cells were incubated for 24, 48 and 72 h with the tested extracts and doxorubicin in order to evaluate their cytotoxicity (after 24 h) and antiproliferative effects (after 48 and 72 h). The results are summarized in the form of bar diagrams in Figure 1.

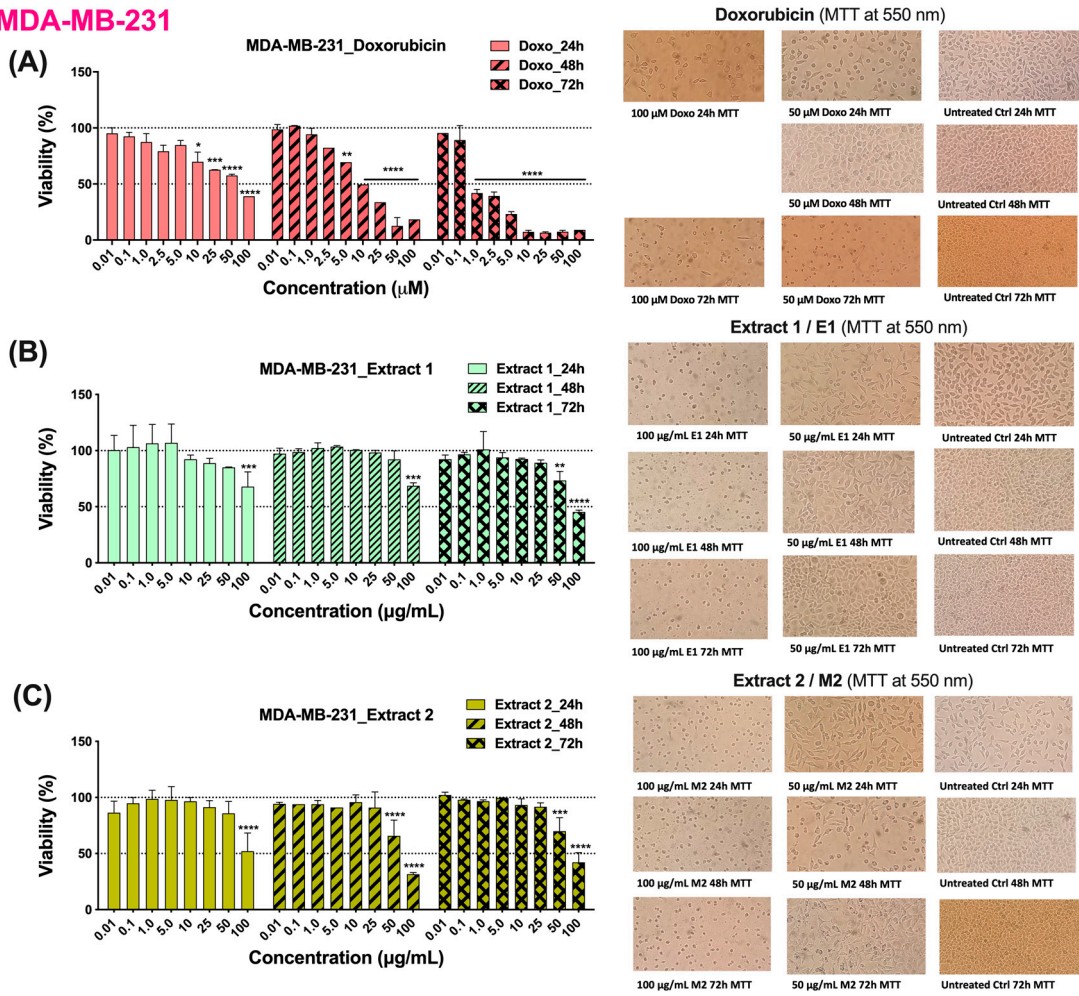

**Figure 1.** Cytotoxicity profile of doxorubicin (**A**), Extract 1/E1 (**B**), and Extract 2/M2 (**C**) measured on MDA-MB-231 cells after 24, 48, and 72 h incubation in the presence of different concentrations of doxorubicin and tested extracts (0.01 to 100 μM for doxorubicin; 0.01 to 100 μg/mL for E1 and M2). Right: Selected pictures of MDA-MB-231 cell culture after 24, 48 and 72 h incubation with doxorubicin and tested extracts (100 μm with 20× magnification). The respective tested concentrations are indicated. The results are the mean % of untreated controls (Ctrl.) ± SD ($n = 3$). One-way ANOVA and Dunnett's multiple comparison test: *, $p < 0.05$; **, $p < 0.01$; ***, $p < 0.001$; ****, $p < 0.0001$ vs. control.

As expected, the reference doxorubicin exhibits moderate (after 24 h treatment at a concentration of 100 μM), significantly improved (after 48 h in the concentration range of 25–100 μM, <50% cell viability) and much higher inhibitory activity (after 72 h treatment in the concentration range of 1.0–100 μM, <50% cell viability) against MDA-MB-231 cell line (Figure 1A). The obtained bar diagrams for the inhibitory activity of both HR extracts on MDA-MD-231 cell growth showed that Extract 1 (E1) exhibits a slight effect after 72 h incubation only at its highest-tested concentration of 100 μg/mL (approx. 48% cell viability, Figure 1B), while Extract 2 (M2) moderately inhibits MDA-MB-231 cells at the same highest-tested concentration of 100 μg/mL after 48 h (about 25% cell viability) and 72 h (about 30–40% cell viability) incubation time (Figure 1C). However, both extracts did not show any cytotoxic effects after 24 h incubation with MDA-MB-231 cells. In this first series of tests, we found that the tested extracts possibly interfere with the emission spectra of the MTT reagent used throughout the experiments. This is probably due to the antioxidative properties of both extracts, which may cause in situ reduction of MTT (550 and 570 nm emission), and, therefore, may lead to a huge deviation of the measured OD values. With

respect to this, it is also recommended to use an additional assay in order to validate these initial data for the cytotoxic and antiproliferative effects of the tested extracts.

### 3.2. Growth Inhibitory Activity of E1 and M2 against HT-29 Cancer Cell

Next, both extracts were tested for their inhibitory activity on human colon adenocarcinoma (HT-29) cell line. The MTT tests were performed in triplicate and, in each experiment, doxorubicin was used as a reference compound. The cells were incubated for 24, 48 and 72 h with the tested extracts and doxorubicin in order to evaluate their cytotoxicity (after 24 h) and antiproliferative effects (after 48 and 72 h). The results are summarized in the form of bar diagrams in Figure 2.

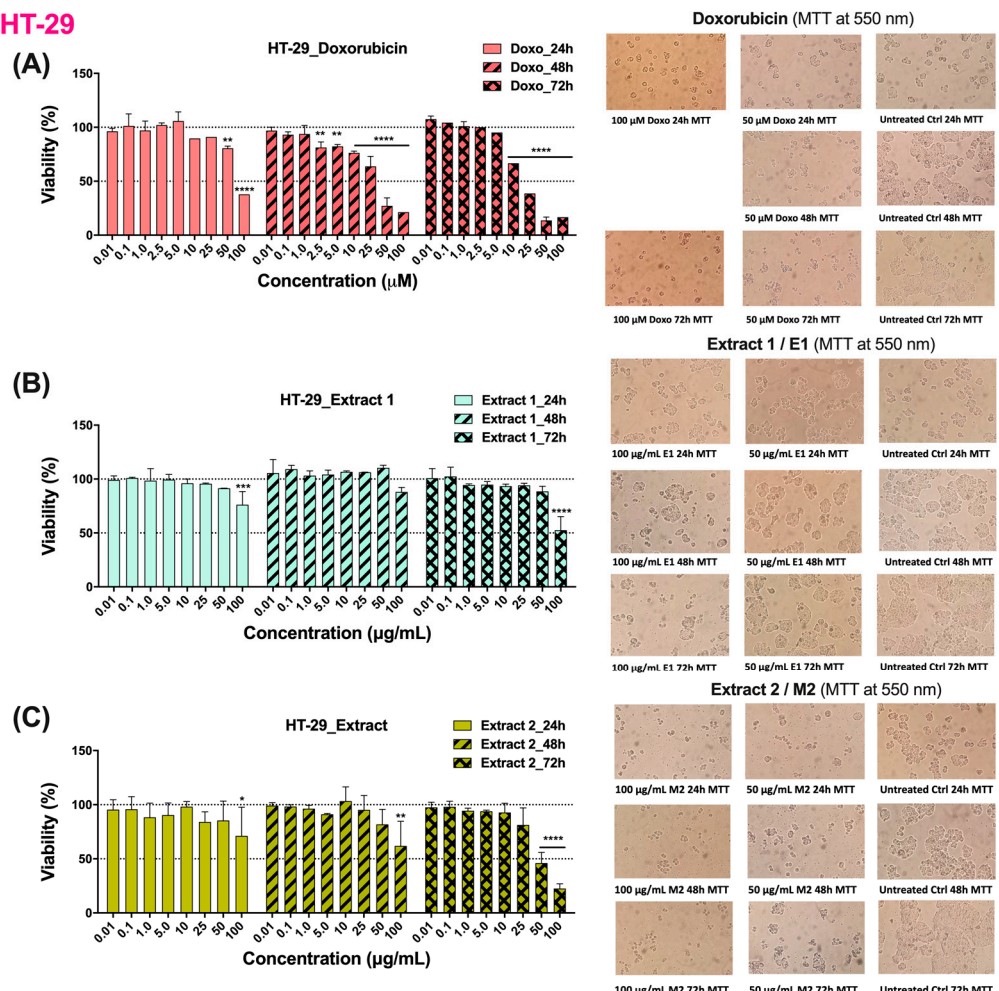

**Figure 2.** Cytotoxicity profile of doxorubicin (**A**), Extract 1/E1 (**B**), and Extract 2/M2 (**C**) measured on human colon adenocarcinoma HT-29 cells after 24, 48, and 72 h incubation in the presence of different concentrations of doxorubicin and tested extracts (0.01 to 100 μM for doxorubicin; 0.01 to 100 μg/mL for Extract 1 and 2). Right: Selected pictures of HT-29 cell culture after 24, 48 and 72 h incubation with doxorubicin and tested extracts (100 μm with 20× magnification). The respective tested concentrations are indicated. The results are the mean % of untreated controls (Ctrl.) ± SD (*n* = 3). One-way ANOVA and Dunnett's multiple comparison test: *, *p* < 0.05; **, *p* < 0.01; ***, *p* < 0.001; ****, *p* < 0.0001 vs. control.

Compared to its inhibitory activity against MDA-MB-231 cancer cells, doxorubicin showed similar cytotoxic effects after 24 h incubation with human colon adenocarcinoma cells (cell viability about 40% at 100 μM), but a slight decrease in antiproliferative effects after 48 h (cell viability about 20–25% at 50 and 100 μM) and 72 h exposure (cell viability about 10–40% at 25, 50 and 100 μM) (Figure 2A). Similar to the effects against MDA-MB-231

cells, Extract 1 (E1) did not exhibit the growth of HT-29 adenocarcinoma cells (Figure 2B), while Extract 2 (M2) decreased its inhibitory activity after 48 h treatment (>50% cell viability at 100 μg/mL) with a slight increase in activity after 72 h exposure with HT-29 cells (<50% cell viability at 50 and 100 μg/mL) (Figure 2C). However, the antiproliferative activity of Extract 2 is comparable to that of doxorubicin at the highest-tested concentration of 100 μM. The cell viability of the HT-29 cells was measured to be 16.7 vs. 22.6% after 72 h treatment with doxorubicin and M2, respectively.

### 3.3. Growth Inhibitory Activity of E1 and M2 against HepG2 Cancer Cell Line

Following our experimental design, both extracts were tested for their ability to inhibit the growth of human hepatocellular carcinoma (HepG2) cell line. The MTT tests were performed in triplicate and, in each experiment, doxorubicin was used as a reference compound. The cells were incubated for 24, 48 and 72 h with the tested extracts and doxorubicin in order to evaluate their cytotoxicity (after 24 h) and antiproliferative effects (after 48 and 72 h). The results are summarized in the form of bar diagrams in Figure 3.

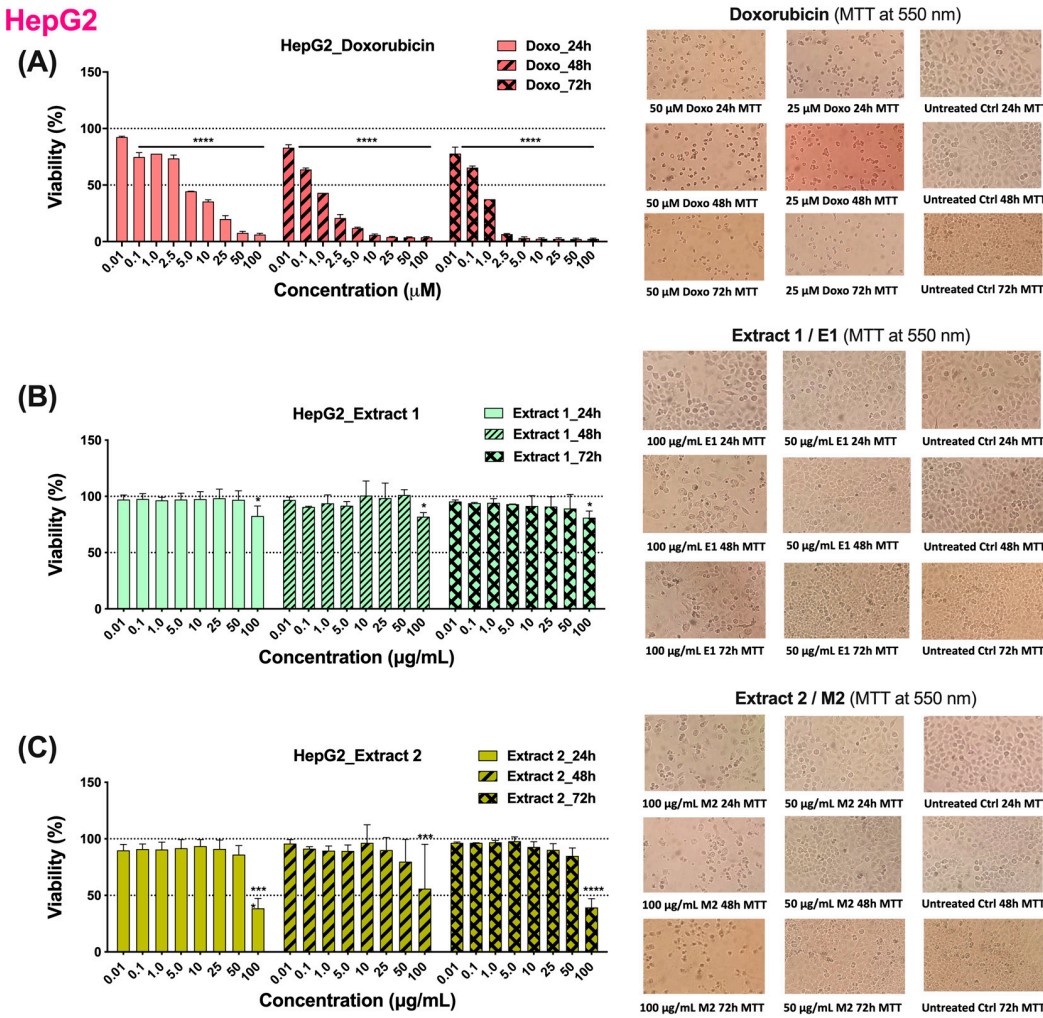

**Figure 3.** Cytotoxicity profile of doxorubicin (**A**), Extract 1/E1 (**B**), and Extract 2/M2 (**C**) measured on human hepatocellular carcinoma HepG2 cells after 24, 48, and 72 h incubation in the presence of different concentrations of doxorubicin and tested extracts (0.01 to 100 μM for doxorubicin; 0.01 to 100 μg/mL for Extract 1 and 2). Right: Selected pictures of HepG2 cell line after 24, 48 and 72 h incubation with doxorubicin and tested extracts (100 μm with 20× magnification). The respective tested concentrations are indicated. The results are the mean % of untreated controls (Ctrl.) ± SD (*n* = 3). One-way ANOVA and Dunnett's multiple comparison test: *, $p < 0.05$; ***, $p < 0.001$; ****, $p < 0.0001$ vs. control.

As illustrated in Figure 3A, doxorubicin showed significant improvement in inhibitory activity against HepG2 when compared to its effects on MDA-MB-231 and HT-29 cancer cell lines. The compound inhibits HepG2 cells after 24 h incubation in the concentration range 5.0–100 μM, and the proliferation of HepG2 cells at concentration ranges 1.0–100 μM (after 48 and 72 h incubation period). The antiproliferative effect of doxorubicin at both time periods is almost equal (Figure 3A). The results of both extracts in terms of their cytotoxicity (after 24 h) and antiproliferation (after 48 and 72 h) showed that Extract 1 (E1) did not inhibit the growth of HepG2, neither after 24 h, nor after 48 or 72 h incubation with HepG2 cells (Figure 3B). The inhibitory activity of Extract 2 (M2) against HepG2 cells is similar to that observed against the other two cancer cell lines (MDA-MB-231 and HT-29) after 48 and 72 h treatment of the respective cancer cells. Compared to its cytotoxic activity against MDA-MB-231 and HT-29 cells, Extract 2 inhibits the growth of HepG2 cells at its highest-tested concentration of 100 μg/mL by more than 60% (Figure 3C). Extract 2 showed similar inhibitory activity after 72 h treatment of MDA-MB-231 at a concentration of 100 μg/mL. Interestingly, inhibitory effects for Extract 2 against HepG2 cells were not observed after a 48 h incubation period, but, as seen from the bar diagram, the measured SD is higher than 20%. Both tested extracts did not show significant improvement in inhibitory activity against HepG2 cancer cells when compared to the strongest cytotoxic and antiproliferative effects of doxorubicin.

### 3.4. Growth Inhibitory Activity of E1 and M2 against 3T3 Cancer Cell Line

In order to evaluate the cytotoxic and antiproliferative effects, both extracts were tested for their ability to inhibit the growth of a non-cancer mouse embryonic fibroblasts 3T3/L1 cell line. The MTT tests were performed in triplicate and, in each experiment, doxorubicin was used as a reference compound. The cells were incubated for 24, 48 and 72 h with the tested extracts and doxorubicin. The results are summarized in the form of bar diagrams in Figure 4.

As indicated in the figure, the reference doxorubicin showed comparable cytotoxic (after 24 h treatment) and antiproliferative effects (after 48 and 72 h treatment) against the non-cancer 3T3 cell line (Figure 4A). After 24 h treatment of 3T3 cells, doxorubicin inhibits the cell growth in the concentration range of 2.5–100 μM; however, the estimated cell viability was between 40 and 50%. In contrast, the inhibitory activity of doxorubicin after 48 and 72 h treatment of 3T3 cells was found to be in the ranges 2.5–100 μM (after 48 h) and 0.5–100 μM (after 72 h). The observed cytotoxic and antiproliferative effects for doxorubicin are comparable to its effects against HepG2. Interestingly, the non-cancer cell line 3T3 was the only one that showed to be sensitive to both tested extracts. In comparison to the determined effects against the tested cancer cell lines (e.g., MDA-MB-231, HT-29, and HepG2), E1 showed slight inhibitory effect on cell growth at its highest-tested concentration of 100 μg/mL after 24, 48 and 72 h incubation with 3T3 cells (Figure 4B). M2 showed similar inhibitory activity against 3T3 cells. It inhibited 3T3 cells at its highest-tested concentration of 100 μg/mL (Figure 4C).

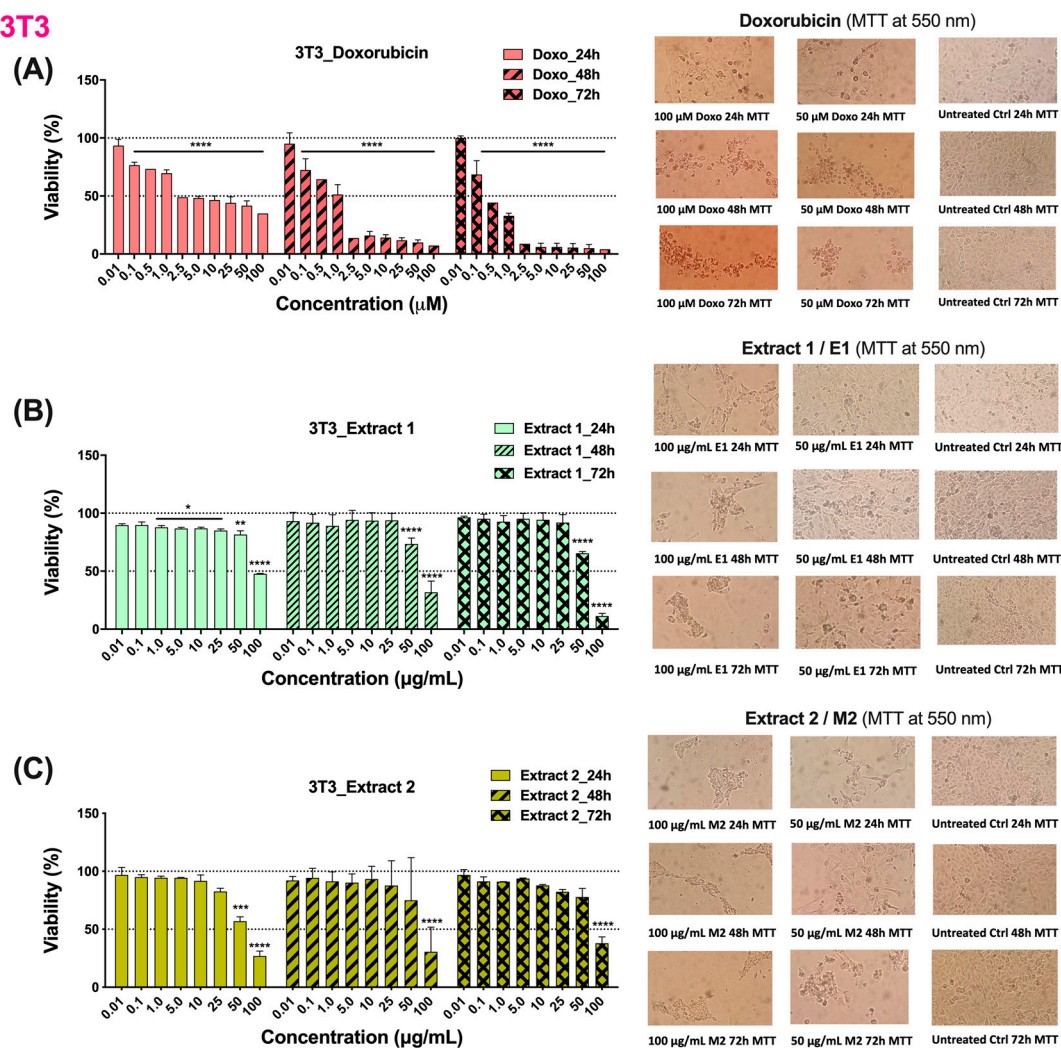

**Figure 4.** Cytotoxicity profile of doxorubicin (**A**), Extract 1 (**B**), and Extract 2 (**C**) measured on mouse embryonic fibroblasts 3T3 cells after 24, 48, and 72 h incubation in the presence of different concentrations of doxorubicin and tested extracts (0.01 to 100 µM for doxorubicin; 0.01 to 100 µg/mL for Extract 1 and 2). Right: Selected pictures of 3T3 cell cells after 24, 48 and 72 h incubation with doxorubicin and tested extracts (100 µm with 20× magnification). The respective tested concentrations are indicated. The results are the mean % of untreated controls (Ctrl.) ± SD (*n* = 3). One-way ANOVA and Dunnett's multiple comparison test: *, $p < 0.05$; **, $p < 0.01$; ***, $p < 0.001$; ****, $p < 0.0001$ vs. control.

## 4. Discussion

The obtained results from all performed MTT tests are summarized in Figure S2. The reference doxorubicin showed inhibitory activity against all investigated cell lines, the cancer cells MDA-MB-231, HT-29, and HT-29, as well as against reference 3T3 cells. The highest inhibitory activity of doxorubicin was measured against HepG2 and 3T3 cells after 48 and 72 h post-treatment (Figure S2, left). The strongest antiproliferative effect for doxorubicin was observed after 72 h incubation with the non-cancer cell line 3T3 (at concentrations between 0.5 and 100 µM). Similar antiproliferative effect for doxorubicin is measured in the concentration range of 1.0–100 µm against HepG2 cancer cells. The lowest inhibitory activity of doxorubicin was assessed against HT-29 cancer cells, independently of the incubation time (at 24, 48, or 72 h).

Compared to doxorubicin, both tested HR extracts showed an overall decrease in inhibitory activity against all cancer (MDA-MB-231, HT-29, and HepG2) and non-cancer (3T3) cell lines. The MTT tests indicated that both extracts did not show comparable dose- and time-dependent (after 24, 48 or 72 h post-treatment) inhibition of cell growth to those

of doxorubicin. However, it can be seen that Extract 1 is not able to inhibit the growth of cancer cell lines (e.g., MDA-MB-231, HT-29, or HepG2), while its inhibitory activity against the non-cancer 3T3 cells is comparable to that observed for Extract 2. Extract 1 was able to inhibit the growth of 3T3 cells after 48 and 72 h post-treatment at a concentration of 100 μg/mL (Figure S2, middle). From both extracts, the HR Extract 2 is the most active one. Extract 2 is able to inhibit the growth of all examined cancer cells (e.g., MDA-MB-231, HT-29, or HepG2) after 72 h incubation of the respective cells with the highest-tested concentration of 100 μg/mL (*cf.* the red arrow in Figure S2, right panel). The strongest effect for Extract 2 is determined after 24, 48 and 72 h against the non-cancer 3T3 cells (at the highest-tested concentration of 100 μg/mL). The cytotoxicity and antiproliferative effects of HR Extract 2 against 3T3 cells are almost similar to the effects of Extract 1. In order to obtain the respective $IC_{50}$ values for doxorubicin (in μM), HR Extract 1 and 2 (in μg/mL), dose-dependent non-linear curves were built (Figure 5).

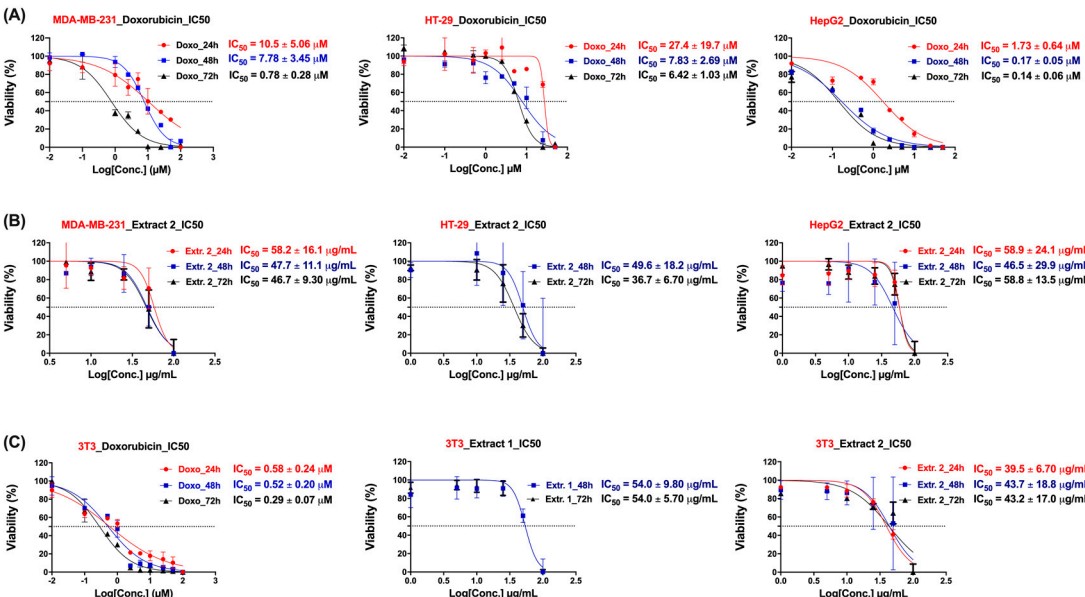

**Figure 5.** Comparison of non-linear inhibitory curves (log[inhibitor] vs. normalized response/ viability in % by variable slope) for doxorubicin (**A**), Extract 2 (**B**), measured on MDA-MB-231, HT-29, and HepG2 cancer cells as well as for doxorubicin, Extract 1 and Extract 2 on 3T3 non-cancer cells (**C**) after 24, 48, and 72 h incubation with different concentrations of doxorubicin and tested extracts (0.01 to 100 μM for doxorubicin; 0.01 to 100 μg/mL for Extract 1 and 2). The respective $IC_{50}$ values (in μM for doxorubicin or μg/mL for Extract 1 and 2) are provided (right). The results are the mean % of untreated controls (Ctrl.) $\pm$ SD (*n* = 3).

The dose-response curves were built after the normalization of transformed values, so that the curves run from 100% (vs. control) down to 0%, depending on cell viability (in % vs. control group) that is measured in MTT assay for each cell line and experiment (e.g., after 24, 48 or 72 h treatment). It can be seen that only one curve was built for Extract 1 since it was evaluated to antiproliferative activity (after 48 and 72 h) only against 3T3 cells. Based on the obtained dose-dependent curves, the inhibitory activity of both tested HR extracts and the reference doxorubicin against three cancer (e.g., MDA-MB-231, HT-29, and HepG2) and one non-cancer cell line (3T3) was determined with higher $IC_{50}$ values for 24 h (cytotoxic effects) compared to 48 and 72 h post-treatment, respectively. The $IC_{50}$ values (expressed in μg/mL) for both investigated HR extracts and doxorubicin are summarized in Table 1.

**Table 1.** Inhibitory potencies (expressed as IC$_{50}$ values) of the reference doxorubicin (Doxo), Extract 1/E1, and Extract 2/M2 after 24, 48 or 72 h treatment of cancer cell lines (MDA-MB-231, HT-29, and HepG2), and the reference non-cancer cell line 3T3.

| Extract | IC$_{50}$ ± SD (µg/mL) [a] | | | | | | | | | | | |
| --- | --- | --- | --- | --- | --- | --- | --- | --- | --- | --- | --- | --- |
| | MDA-MB-231 | | | HT-29 | | | HepG2 | | | 3T3 | | |
| | 24 h | 48 h | 72 h | 24 h | 48 h | 72 h | 24 h | 48 h | 72 h | 24 h | 48 h | 72 h |
| **Doxo** [b] | 10.5 ± 5.06 | 7.78 ± 3.45 | 0.78 ± 0.28 | 27.4 ± 19.7 | 7.83 ± 2.69 | 6.42 ± 1.03 | 1.73 ± 0.64 | 0.17 ± 0.05 | 0.14 ± 0.06 | 0.58 ± 0.24 | 0.52 ± 0.20 | 0.29 ± 0.07 |
| **E1** | >100 | >100 | >100 | >100 | >100 | >100 | >100 | >100 | >100 | >100 | 54.0 ± 9.8 | 54.0 ± 5.7 |
| **M2** | 58.2 ± 16.1 | 47.7 ± 11.1 | 46.7 ± 9.3 | >100 | 49.6 ± 18.2 | 36.7 ± 6.7 | 58.9 ± 24.1 | 46.5 ± 29.9 | 58.8 ± 13.5 | 39.5 ± 6.7 | 43.7 ± 18.8 | 43.2 ± 17.0 |

(*a*) Data are from three independent experiments (*n* = 3). (*b*) Reference compound doxorubicin (Doxo).

In regard to HR extract E1, the IC$_{50}$ values at 48 and 72 h were determined to be equal to 54.0 µg/mL (50% inhibition of cell growth) for 3T3 cells, whereas, for all other cell lines, they could not be estimated. Therefore, the IC$_{50}$ values for E1 for all cancer lines are higher than the highest-tested concentration of 100 µg/mL (IC$_{50}$ > 100 µg/mL, cf. Table 1). In contrast, the HR extract M2 inhibits the cell growth of all cancer cell lines by about 50–60% (with the exception of HT-29 cells at 24 h), and by about 40% of 3T3 non-cancer cells. Therefore, the IC$_{50}$ values for M2 at 24 h were determined to be higher (approx. 58 µg/mL for MDA-MB-231, approx. 59 µg/mL for HepG2, and approx. 40 µg/mL for 3T3) than those measured at 48 or 72 h post-treatment, showing that M2 exhibits stronger antiproliferative effects than cytotoxic activity (increasing by about 10–15%, on average). Moreover, M2 showed increased cytotoxicity and antiproliferative affects compared to E1, which was not able to inhibit the growth of all cancer cell lines.

The highest cytotoxic effect at 24 h for M2 was determined against 3T3 cells (approx. 40 µg/mL), while the strongest antiproliferative effect was estimated at 72 h post-treatment of HT-29 cancer cells (approx. 37 µg/mL). Overall, it can be concluded that M2 moderately inhibits MDA-MB-231, HT-29, HepG2, and 3T3 growth with IC$_{50}$ values in the range of 40–59 µg/mL, whereas Extract 1 (with the exception of 3T3 cells) did not exhibit cytotoxic (at 24 h) or antiproliferative effects (at 48 and 72 h) against all investigated cancer cells (Table 1). However, the cytotoxicity and antiproliferative activity of M2 are not comparable to the effects observed for the reference doxorubicin. The strongest IC$_{50}$ values for doxorubicin were determined to be 0.17 and 0.14 µg/mL against HepG2 cancer cells at 48 or 72 h, respectively (*cf.* Table 1). In general, depending on the tested cells, doxorubicin was evaluated to be between 10- and 300-fold more active than M2 at the different evaluated time points (e.g., 24, 48, or 72 h post-treatment).

## 5. Conclusions

In conclusion, the myconoside-enriched HR plant Extract 2 (M2) showed increased inhibitory activity (cytotoxicity and antiproliferative effects) compared to the HR plant Extract 1. Moreover, the plant Extract 2 showed a significant increase in cytotoxicity (at 24 h) and antiproliferative activity (at 48 and 72 h post-treatment) at its highest-tested concentration of 100 µg/mL compared to plant Extract 1. As obtained herein, both plant Extracts 1 and 2 inhibited the growth of the non-cancer cell line 3T3 at their highest-tested concentration of 100 µg/mL. The reference compound doxorubicin, used in this study as a reference compound, showed strong cytotoxic (at 24 h) and antiproliferative effects (at 48 and 72 h post-treatment) against all tested cell lines (cancer and non-cancer cells) being between 10- and 300-fold more active than the most active plant Extract 2. Since it is considered a skin-protecting cosmetic product, the plant Extract 2 can also be considered for further time- and dose-dependent experiments in order to evaluate its cytotoxicity and antiproliferative effects against a panel of metastatic melanoma cancer cell lines, e.g., SK-MEL-3, SH-4, SK-MEL-24, and other cells.

**Supplementary Materials:** The following supporting information can be downloaded at: https://www.mdpi.com/article/10.3390/cosmetics11020046/s1, Figure S1: (A) The plant *Habelea rhodopensis* Friv. (B) Chemical structures and IUPAC names of the main constituents of *H. rhodopensis* Friv. myconoside and paucifloside.; Table S1: Transformation of doxorubicin concentrations (from μM to μg/mL); Figure S2: Comparison of the cytotoxicity profile for doxorubicin (left), Extract 1 (middle), and Extract 2 (right) measured on MDA-MB-231, HT-29, and HepG2 cancer cells, as well as on 3T3 non-cancer cells after 24, 48, and 72 h exposure to different concentrations of compounds (0.01 to 100 μM for doxorubicin; 0.01 to 100 μg/mL for Extract 1 and 2). The respective substance concentrations are indicated. For simplification, the red arrow (right) showed the determined trend in inhibitory activity for extract 2, determined at its highest tested concentration of 100 μg/mL. Untreated (control, Ctrl.) cells were used as a positive control. The results are expressed as the mean % of untreated controls $\pm$ SD ($n = 3$). Statistical analysis was performed by one-way ANOVA and Dunnett's multiple comparison test. *, $p < 0.1$; **, $p < 0.01$; ***, $p < 0.001$; ****, $p < 0.0001$ *vs.* control.

**Author Contributions:** Conceptualization, collection and assembly of data, data analysis and interpretation, N.T.T.; methodology, collection and assembly of data, data analysis and interpretation M.I.P., M.G.G. and A.A.B.; software, N.T.T. and M.G.G.; validation of data, M.I.P., M.G.G. and A.A.B.; investigation, M.I.P., M.G.G. and A.A.B.; resources, N.T.T. and A.P.; data curation, M.I.P.; writing—original draft preparation, N.T.T.; writing—review and editing, M.I.P., A.P. and N.T.T.; visualization, M.I.P. and M.G.G.; supervision, N.T.T.; project administration, A.P.; funding acquisition, N.T.T and A.P. All authors have read and agreed to the published version of the manuscript.

**Funding:** This research received no external funding.

**Institutional Review Board Statement:** Not applicable.

**Informed Consent Statement:** Not applicable.

**Data Availability Statement:** Data are contained within the article and the Supplementary Materials.

**Conflicts of Interest:** The authors declare no conflicts of interest. Atanas Pavlov is employed by the company Innova BM Ltd. The remaining authors declare that the research was conducted in the absence of any commercial or financial relationships that could be construed as potential conflicts of interest. The authors declare that the company Innova BM Ltd. provided the materials. They were not involved in the study design, collection, analysis, interpretation of data, the writing of this article or the decision to submit it for publication.

## Abbreviations

| | |
|---|---|
| ANOVA | analysis of variance |
| DMSO | *N,N*-dimethyl sulfoxide |
| Doxo | doxorubicin |
| HR | *Habelea rhodopensis* Friv. |
| HREs | *H. rhodopensis* Friv. extracts |
| IUPAC | International Union of Pure and Applied Chemistry |
| MTT | 3-(4,5-dimethylthiazol-2-yl)-2,5-diphenyltetrazolium bromide |
| SD | Standard Deviation |
| (U)HPLC-MS | (ultra-)High Performance Liquid Chromatography-Mass Spectrometry. |

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
