# Peer review of "In Vitro Investigation of the Cytotoxic and Antiproliferative Effects of Haberlea rhodopensis Total Extract: A Comparative Study"

_cosmetics, doi:10.3390/cosmetics11020046_

Round 1

Reviewer 1 Report

Comments and Suggestions for Authors

The work entitled “In vitro investigation of the cytotoxic and antiproliferative effects of Haberlea rhodopensis total extract: A comparative study” submitted by Pavlov A. and collaborators describes the evaluation of Haberlea rhodopensis extracts, as cytotoxic and antiproliferative compared with doxorubicin. Since extracts of this species have been used as an ingredient in cosmetics, the work falls within the “toxicological studies of cosmetic products” section of MDPI Cosmetics.

The work is well executed from an experimental point of view and provides results that may be of value in relation to the use of extracts of this species in cosmetic preparations. Although it would have been very interesting for the authors to carry out the evaluation of the so-called extract 2 in a panel of metastatic melanoma cancer cell lines as established in the conclusions, I consider that the work has sufficient relevance to be published in Cosmetics and may be of interest to a specialized public.

The paper is well presented, and the discussion and conclusions sections agree with the experimental results.

Author Response

Comments and Suggestions for Authors

“The work entitled “In vitro investigation of the cytotoxic and antiproliferative effects of Haberlea rhodopensis total extract: A comparative study” submitted by Pavlov A. and collaborators describes the evaluation of Haberlea rhodopensis extracts, as cytotoxic and antiproliferative compared with doxorubicin. Since extracts of this species have been used as an ingredient in cosmetics, the work falls within the “toxicological studies of cosmetic products” section of MDPI Cosmetics.

The work is well executed from an experimental point of view and provides results that may be of value in relation to the use of extracts of this species in cosmetic preparations. Although it would have been very interesting for the authors to carry out the evaluation of the so-called extract 2 in a panel of metastatic melanoma cancer cell lines as established in the conclusions, I consider that the work has sufficient relevance to be published in Cosmetics and may be of interest to a specialized public.

The paper is well presented, and the discussion and conclusions sections agree with the experimental results.”

Authors’ reply:

We are grateful for this comment and appreciate the Reviewer’s opinion regarding our manuscript and his/her important suggestion. In fact, we plan the testing of extract 2 against melanoma cell lines as a part of our further work, which will also include additional experiments only with extract 2.

Reviewer 2 Report

Comments and Suggestions for Authors

The manuscript proposed by Peeva and co-workers is focused on in vitro investigation of the cytotoxic and antiproliferative effects of Haberlea rhodopensis Friv. extracts against reference anticancer, non-selective compound doxorubicin. This is an interesting work. As a consequence, I recommend publication of this potentially useful work with the minor revision proposed herein. Some typos mistakes are present in the manuscript, for instance and not exhaustively, please see the observations on the manuscript.

Comments on the Quality of English Language

The manuscript is generally well written in English. However, there are some flaws that may need to be addressed.

Author Response

Comments and Suggestions for Authors

“The manuscript proposed by Peeva and co-workers is focused on in vitro investigation of the cytotoxic and antiproliferative effects of Haberlea rhodopensis Friv. extracts against reference anticancer, non-selective compound doxorubicin. This is an interesting work. As a consequence, I recommend publication of this potentially useful work with the minor revision proposed herein. Some typos mistakes are present in the manuscript, for instance and not exhaustively, please see the observations on the manuscript.”

Authors’ reply:

We thank the Reviewer for the time and effort invested to review our work and for all improvement-recommendations. We appreciate the enclosed pdf-revision file that clearly indicates reviewer’s remarks and suggestions for improving the manuscript. We agree with the reviewer’s comments and implemented his/her remarks and suggestions into the revised manuscript.

Comments on the Quality of English Language

“The manuscript is generally well written in English. However, there are some flaws that may need to be addressed”.

Authors’ reply:

We thank the Reviewer for his/her comment. Therefore, we improved the text and corrected such flaws throughout the manuscript.

Reviewer 3 Report

Comments and Suggestions for Authors

The manuscript entitled “In vitro investigation of the cytotoxic and antiproliferative effects of Haberlea rhodopensis total extract: A comparative study” falls within the scope of the Journal. However, this reviewer has the following comments for the manuscript.

Major comments:

- What is the vehicle used to solubilise the E1 and M2 extracts? Authors should indicate the vehicle in the materials and methods section, stating that the vehicle was used for the control condition in the cytotoxicity and cell proliferation assays.

- Do authors have data on the cell viability and anti-proliferative effects of the compounds even at the 4-hour time point? The authors should include this data (also in the supplementary data).

- Considering the role of inflammation in tumour processes, from my point of view in addition to the effect of the substances on the 3t3 fibroblast cell line, authors should also perform the same assays on macrophage cell lines (e.g. J774 or RAW murine macrophage cell line).

- Authors should insert a graphical abstract that summarizes the contents of the article in a concise form in order to capture the attention of the readership.

Minor comments:

- Authors should report the keywords in alphabetical order.

- Authors should insert an abbreviation section. The words for which is specified an abbreviation should be written in full the first time they are mentioned.

- The English language has to be extensively revised.

- Authors should improve the formal aspects of the manuscript.

Comments on the Quality of English Language

- The English language has to be extensively revised.

Author Response

Comments and Suggestions for Authors

The manuscript entitled “In vitro investigation of the cytotoxic and antiproliferative effects of Haberlea rhodopensis total extract: A comparative study” falls within the scope of the Journal. However, this reviewer has the following comments for the manuscript.

Major comments:

- “What is the vehicle used to solubilise the E1 and M2 extracts? Authors should indicate the vehicle in the materials and methods section, stating that the vehicle was used for the control condition in the cytotoxicity and cell proliferation assays”.

Authors’ reply:

We thank the Reviewer for his/her fruitful comment and these important remarks. The vehicle used to solubilise the E1 and M2 extracts is ddH2O, while the doxorubicin was dissolved in DMSO. This is already indicated in the materials and methods section in the subhead “preparation of the stock solution”. But the reviewer is right that it is appropriate to further emphasize that both solvents were used as negative controls. The solvents were used at concentrations that didn’t influence cell viability, and that now is properly indicated in the materials and methods section.

- “Do authors have data on the cell viability and anti-proliferative effects of the compounds even at the 4-hour time point? The authors should include this data (also in the supplementary data).”

Authors’ reply:

We thank the Reviewer for this comment. In fact, we did not perform MTT test after a 4-hour treatment with the investigated extracts. The most common time points for performance of MTT assay are after 24-hour treatment (when cytotoxicity is expected) or 72-hour treatment (when anti-proliferative effect is expected). We believe that the 4-hour treatment is a time period that is sufficient to initiate a process such as apoptosis for example, but these early phases and events are better detected by other more precise methods, such as FACS analysis. MTT assay is usually used as a fast-screening test for cytotoxicity and/or antiproliferation. We are also afraid that within this short 4-hour period the metabolic activity of the cells would not be change and the possibility of false positive/negative results is high. But we are still grateful for the reviewer’s suggestion and we may try in further experiments and also under proper conditions (or assays).

- Considering the role of inflammation in tumour processes, from my point of view in addition to the effect of the substances on the 3t3 fibroblast cell line, authors should also perform the same assays on macrophage cell lines (e.g. J774 or RAW murine macrophage cell line).

Authors’ reply:

We appreciate the Reviewer’s recommendation. Our current study represents an initial screening of the two extracts in order to gain insight into their potential for further investigations and applications. In this context, the reviewer’s suggestion is very valuable as it not only refers to possible further experiments but also addresses the problem of inflammation which is sometimes overlooked. The use of macrophage cell line would make the study of the full potential of these extracts more comprehensive.

- Authors should insert a graphical abstract that summarizes the contents of the article in a concise form in order to capture the attention of the readership.

Authors’ reply:

We appreciate the Reviewer’s recommendation. We now included a Graphical abstract as a separate file that summarizes the content of the article.

Minor comments:

- Authors should report the keywords in alphabetical order.

Authors’ reply:

Keywords are now reported in alphabetical order.

- Authors should insert an abbreviation section. The words for which is specified an abbreviation should be written in full the first time they are mentioned.

Authors’ reply:

We appreciate the Reviewer’s recommendation. Therefore, an abbreviation section is now inserted in in the revised manuscript.

- The English language has to be extensively revised.

Authors’ reply:

We revised the English language by the help of a native speaker.

- Authors should improve the formal aspects of the manuscript.

Authors’ reply:

The formal aspects of the manuscript are now improved.

Round 2

Reviewer 3 Report

Comments and Suggestions for Authors

I have read the revised version of the manuscript named "In vitro investigation of the cytotoxic and antiproliferative effects of Haberlea rhodopensis total extract: A comparative study". The authors have made revisions to this article in accordance with the suggestions of reviewers. I think that manuscript is worth publishing in Cosmetics.